# Music Generation System for Adversarial Training Based on Deep Learning

**Jun Min [1], Zhaoqi Liu [1], Lei Wang [1,\*], Dongyang Li [1], Maoqing Zhang [1] and Yantai Huang [2]**

1 College of Electronics and Information Engineering, TongJi University, Shanghai 201804, China
2 College of Automation and Electrical Engineering, Zhejiang University of Science and Technology, Hangzhou 310023, China
\* Correspondence: wanglei@tongji.edu.cn

**Abstract:** With the rapid development of artificial intelligence, the application of this new technology to music generation has attracted more attention and achieved gratifying results. This study proposes a method for combining the transformer deep-learning model with generative adversarial networks (GANs) to explore a more competitive music generation algorithm. The idea of text generation in natural language processing (NLP) was used for reference, and a unique loss function was designed for the model. The training process solves the problem of a nondifferentiable gradient in generating music. Compared with the problem that LSTM cannot deal with long sequence music, the model based on transformer and GANs can extract the relationship in the notes of long sequence music samples and learn the rules of music composition well. At the same time, the optimized transformer and GANs model has obvious advantages in the complexity of the system and the accuracy of generating notes.

**Keywords:** artificial intelligence (AI); music generation; natural language processing; transformer; GANs

## 1. Introduction

Computer-based music generation can be traced as far back as the 1950s. The Russian mathematician Markov proposed the Markov model in 1906, which processed and predicted serial data. In 1955, Olson et al. [1] proposed using the Markov model to compose music for the first time, utilizing intelligent computer algorithms that calculate probabilities for the next note. In his research, Olson described the principal modules needed for analog electronic music synthesizers, providing a creative method through formal methods. Similar algorithms based on this approach are directed by a control program that must follow mathematical instructions in a fixed order. By the 1980s, various models had appeared offering grammar-based [2] and rule-based music generation [3]. Steedman et al. [2] referred to the grammatical rules of language-generation systems to build a set of music generation systems with pure grammar and rules based on music composition theory, such as harmony, polyphony, musical form, and orchestration [3,4]. These models could generate different styles of music but lacked the ability of generalization, so the rule-based definitions had to be completed manually. Compared with deep-learning models, these methods suffered from poor function and universality. However, this basic research provided ideas for optimizing a deep learning algorithm [5].

With the advancement of deep learning techniques, many researchers have recently tried to apply deep neural networks to creating music. These deep learning techniques usually have neural network architectures, which have performed well in computer vision and natural language programming (NLP). Generating music and other artistic content using deep learning is a growing area of research. Generating structured music, analyzing the quality of generated music, and building an interaction model are the extended problems that need to be solved. Some deep learning methods are autoregressive [5]. This

new model tries to generate longer sequences by obtaining the information of past time steps, which is similar to music generation by humans. Convolutional neural networks (CNNs) [6] are a basic model commonly used in music generation. In the late 1990s, the CNN prototype [7] was proposed, but it was not until AlexNet [8] in 2013 that CNNs as we know them were recognized. Based on this, Google established the deep mind artificial intelligence laboratory in London to develop WaveNet [9], one of the most successful CNN music generation applications. The recurrent neural network (RNN) has become the most popular processing model for typical serialized data such as music.

Considering the similarity between music and language, some language generation models can be converted to generating music. This approach represents note embedding by vectors. Embedding [10] was a concept initially used in NLP. Its principle was to obtain the vector representation of each word, note, or chord through training a neural network with a corpus and then use the result as the input to the network for downstream tasks. In music, each major or minor chord has three notes as the tonic of the mode. In other words, the whole song surrounds the tonic and its related chords [11,12]. In addition to this obvious relationship between notes, there are also invisible relationships in music [13], corresponding to anaphora in language. This is why it is important to choose a transformer as the core model for music generation. In addition, GANs are widely used in imaging, language, and music by training the discriminator and generator in the network to optimize them, ensuring the final authenticity of the generated results [14]. GANs are now coming into use to generate music sequences [15]. A critical difficulty to consider is that the generated sequence must be fed into a discriminator because of the network architecture of the generating countermeasure network. Therefore, this study proposes a new model structure that combines a transformer [16] and GANs [17] to create music. The study also presents a unique loss function to enable the system to learn and update the parameters in the two gradient descent directions of "real music" and the target sequence [18].

The contributions of this study are as follows:

- Establishing a new music generation system that combines the transformer and GANs. Proposing a unique loss function for the proposed model to learn from the descending direction of "real music" and the target sequence and to update the parameters over time.
- Improving the input and output structures of the discriminator and the generator and solving the problems of gradient non-differentiability and mode collapse in the discriminator.
- Applying the vocabulary matching method to perfect the intricate melody generated in the time domain and generate a real and controllable long-term structure.
- Presenting a relatively objective suggestion to evaluate music based on Euler's music evaluation mechanism.

The rest of the study is organized as follows: Section 2 introduces related work on music generation. Section 3 puts forward the method proposed in this study and introduces the construction and training process of the model. Section 4 presents the results of the experiment.

## 2. Related Work

Long short-term memory (LSTM) developed rapidly and has been applied extensively in music processing and the music generation field. Through LSTM, automatically generated music can achieve high-quality, high-fidelity, and high-definition music effects. Ycart et al. [19] and Sheykhivand et al. [20] both used LSTM as the cornerstone neural network to realize music generation. Borodin et al. [21] proposed a multi-channel data-processing method for a chord using the many-hot encoding method. The input was optimized from single-note encoding to a multi-dimensional representation vector. The next note combination was predicted by LSTM, which enriched the production. Chen et al. [22] combined LSTM with chaos theory to optimize the tone shift in music without deformation, reduce the amount of calculation, and optimize the training efficiency. Lehner et al. [23]

combined LSTM with a restricted Boltzmann machine (RBM) [24]. These techniques and other probabilistic methods are combined with deep neural networks to help people better make music. However, LSTM methods have failed to generate long-term sequences. The authors of [25] proposed combining biaxial LSTM with GANs, which improved music quality significantly over ordinary LSTM. Many experiments have found that these models only allow the network to learn the relationship between note characteristics from actual music data; they did not learn harmony from the music as a whole or the rules composers need to follow.

Language and music have similar characteristics [26]. In natural language processing, Google put forward the transformer [27] in 2017, which used a self-attention mechanism as its main component and became a state-of-the art (SOTA) method in many projects. The transformer was a typical encoder-decoder model (i.e., a sequence-to-sequence (seq2seq) model) [28] mainly used for generating scenarios such as question-answering systems and machine translation. Given the similarity between language and music, Google applied the transformer to music generation in 2018 [29]. Because of its low memory density, the transformer could generate longer, more coherent music. However, the music transformer was imperfect and had too many redundant and sparse musical notes. Based on the transformer decoder, OpenAI proposed the second-generation Generative Pre-Training (GPT) model [30] in 2019. GPT-3 was considered a dangerous machine learning model due to its high intelligence. In April of the same year, the music generation system MuseNet appeared, based on the GPT-2 [31]. It could generate works of any genre and style, and even any composer and pianist. The generated music could be confused with the official versions. In 2020, Jin et al. [32] proposed a new scheme based on combining the transformer and GPT. Since then, artificial intelligence composition has reached a more mature stage. However, the quality of music generated by these methods has not reached an acceptable level because the neural network cannot understand the complexities of the language of music. The information in the notes needs to be transmitted to the system as part of the input, for example by marking. Many experiments have found that these models only allowed a network to learn the relationship between note characteristics from music data; they did not follow the rules of the music and composition.

## 3. Proposed Method

The first problem is generating note sequences and exploring the conversion relationship. In note conversion, the most commonly used method is to convert notes into one-hot encoding, but because the sample involves many notes, the digital matrix generated by one-hot encoding is sparse. Although one-hot encoding can reduce the dimensionality of the input model, the predicted label probability distribution cannot be directly transformed accurately into notes, so the sorted note sequence is digitally coded according to the occurrence order of the notes. Although this encoding method ignores the original relationship between notes (e.g., the relationship between tone C and chord C-E-G), the model structure of the transformer handles the note ID input by first going through a trainable embedding layer in the system to find a reasonable mapping and converting the note ID into a vector containing the note relationship. In addition, owing to the different lengths of music singles, ranging from dozens of notes to thousands of notes, the seq_length parameter is set to make the input sequence and output sequence into two equal continuous sequences. For example, the first ten notes of a song are converted as follows:

[6.11, E-5, E5, F#4, B3, E-5, B4, 2.6, E-5, 11.3.6].

Among them, "6.11" is a polyphonic chord divided by a period, and the rest are single notes. There are more than 1500 different notes or chords in 2786 samples of music in the GiantMIDI-Piano dataset. To reduce the difficulty of model prediction, all notes are sorted according to their frequency of occurrence, and the notes that occurred less frequently than the set threshold are marked [unk]. The process for note feature extraction is shown in Figure 1.

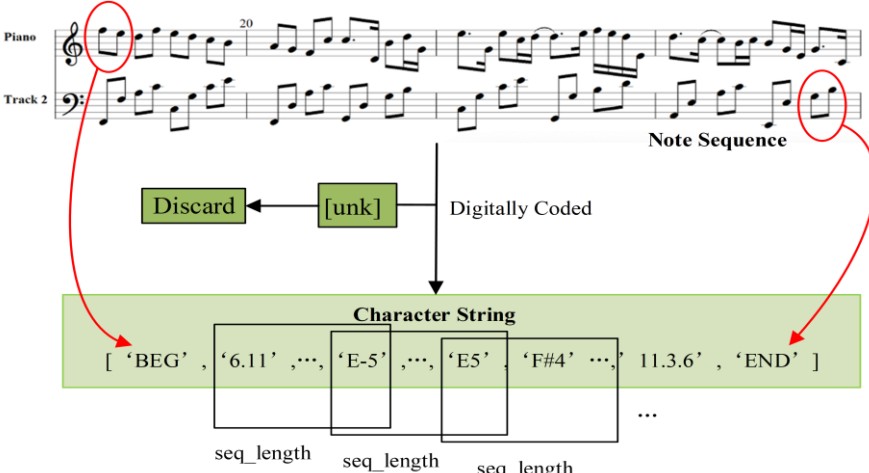

**Figure 1.** The process for note feature extraction.

In the recommended method, the LSTM structure in the traditional sequence-to-sequence (seq2seq) structure is replaced by relying on the attention mechanism to capture the relationship between the input and output notes. Using the attention mechanism, which was proposed by the team of Bengio in 2014 [33], overcomes the shortcomings of sequential calculation of the cyclic neural network (CNN). The system has better parallelism and improves the training efficiency.

In position encoding, different calculation methods are used for note units in odd and even positions. For note units in even positions, Equation (1) is used:

$$PE_{(pos,\ 2i)} = \sin\left(\frac{pos}{10000^{\frac{2i}{d_{model}}}}\right). \tag{1}$$

For note units in odd positions, Equation (2) is used:

$$PE_{(pos,\ 2i+1)} = \cos\left(\frac{pos}{10000^{\frac{2i}{d_{model}}}}\right) \tag{2}$$

where *pos* represents the note, and *i* is a representation dimension, with each dimension corresponding to a sine wave. The position encoding of this method can make the unit $PE_{pos+k}$ at any position be represented by a linear function of the first unit $PE_{pos}$, to obtain the relative position of the whole sequence. The representation method of a trigonometric function is used to make the model infer a sequence with a longer length than that encountered in the training process. Because of the periodicity of sine and cosine, for the fixed-length note sequences, the value of *PE* in the position *pos+n* can be expressed as a linear change. In this way, it will be convenient for the model to learn the relative position relationships between each note.

The encoder is composed of *n* identical subcoding layers, and each encoding layer is composed of two sublayers. The first layer is a multiheaded self-attention mechanism composed of a multihead attention mechanism and a scaled dot-product attention unit. The second layer is a fully connected feed-forward network. The outputs of the two sublayers are connected by residuals, and each layer is normalized to avoid gradient disappearance. The output of each sublayer is $LayerNorm(x + Sublayer(x))$, where $Sublayer(x)$ is the function realized by a sublayer. In the self-attention layer of the encoder, all queries, keys, and values come from the output of the previous layer in the encoder, expressed as the three vectors *Q*, *K*, and *V*, respectively. The calculation formula is as follows:

$$Attention(Q, K, V) = softmax\left(\frac{QK^T}{\sqrt{d_k}}\right)V \tag{3}$$

where $d_k$ is the dimension of the input vector. There are two common attention functions: one of them is additive attention, and the other is dot-product attention. The attention mechanism used here is dot-product attention. This attention mechanism is not only faster than additive attention but also saves more space. The weight of *Attention* is calculated according to $Q$ and $K$, and $d_k$ needs to be scaled; otherwise, when the value of dot-multiplication is too large, the gradient that is calculated by the function *softmax* will be very small, and this is not conducive to backpropagation.

Multihead attention is composed of $h$ self-attention. The calculation formula is shown in Equation (4):

$$MultiHead(Q, K, V) = Concat(head_1, \ldots, head_h)W^O \tag{4}$$

where

$$head_i = Attention\left(QW_i^Q, KW_i^K, VW_i^V\right) \tag{5}$$

Because self-attention only learns from one perspective, it may be biased. Therefore, $h$ different weight combinations are designed. Before calculating *Attention*, $Q$, $K$, and $V$ are linearly transformed with the above weight combination, respectively. The weight of $h$ angles of *Attention* is spliced, and linear transformation is performed with a new weight matrix to get the final output. Multihead attention splits the $Q$, $K$, $V$ vector of the note unit into $h$ word vectors with $d_{model}/h$ dimensionality for self-attention calculation; splices the operation results; merges and adjusts them with the full connection layer; and then outputs the result. The decoder structure is roughly the same as that of the encoder and is composed of $N$ subdecoders. The difference is that the $Q$ and $K$ vectors in the multihead attention input of the decoder come from the encoder. To obtain the relationship between units in the note sequence from data learning and training, the dot product of $Q$ and key $K$, formed by self-attention in the encoding process, becomes the weight of $V$ in the understanding code process. Compared with the encoder layer, a masked multihead attention unit is added to each subdecoder layer, because in generating note sequences, prediction of the next note unit needs to be performed after the prediction of the current note unit. Otherwise, the model is equivalent to directly seeing the question's answer before learning, so the training is meaningless.

A frame diagram of the whole model is shown in Figure 2.

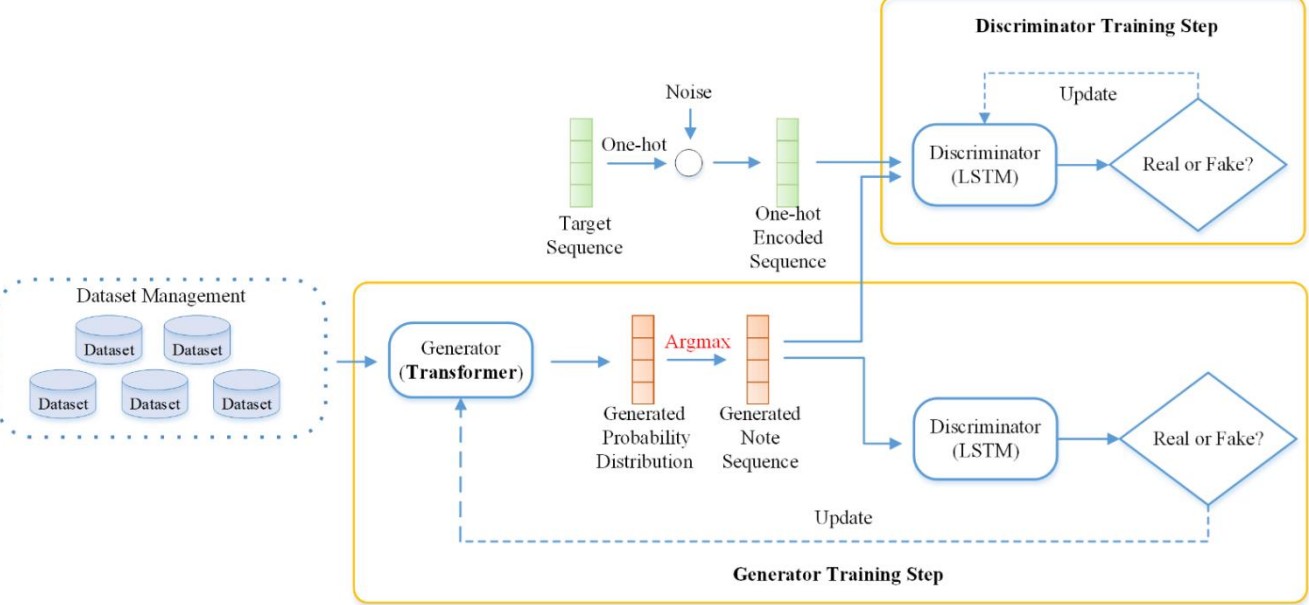

**Figure 2.** The structure of the music generation system.

This system performs alternating confrontation training on generator G and discriminator D in one cycle. In each cycle, the discriminator is trained first. The discrimination model comprises bidirectional long short-term memory (BLSTM), with two layers of 256 units each. The input of the discrimination model is a dataset composed of the real target sequence in the current batch and the sequence generated by the generation model. The gradient descent method is used for training when solving for the gradients of discriminator parameters.

After sequence acquisition, two continuous fixed-length note sequences can be obtained, *input_Seq* and *tar_Seq*. Next, "[BEG]" is added at the beginning of each note sequence, and "[END]" is added at the end of the note sequence; then, the label encoder encodes its ID. There is a trainable embedded layer in the input layer of the model. The mapping relationship between the input and the embedded unit is learned online during the training process.

The converted note vector is multiplied by the position code and input into the transformer encoder composed of an eight-layer subencoder. At this time, the output of the generated model is the probability distribution of dimension [*BatchSize*, *SeqLength*, *NumNotes*], that is, the value in the last dimension of the tensor is the probability of the occurrence of the note corresponding to the digital subscript. With the updated warm-up learning rate, the Adam optimizer is used to train the generator. The calculation formula of this process is as follows:

$$LearningRate = d_{model}^{-0.5} * min\left(StepNum^{-0.5},\ StepNum * WarmupSteps^{-1.5}\right) \qquad (6)$$

where *LearningRate* is the probability distribution of notes, $d_{model}$ is the dimension of the model input vector, and *StepNum* is the number of steps in the current workout. The reason for using the warm-up learning rate strategy is that the compensation mechanism needs to be combined at an early stage of system training. A fast update rate is used so that the model learns the parameter characteristics of notes quickly. In the middle and late stages of training, the learning rate is slowly reduced so that the model can better learn the detailed characteristics of the note distribution. Combined with GANs, the prediction sequence of notes is input to the discriminator to determine whether it is sampled from the dataset or generated. Equation (7) is the optimization function of GANs:

$$\min_{G}\max_{D} V(G, D) = \min_{G}\max_{D} \mathbb{E}_{x \sim p_{data}}[\log D(x)] + \mathbb{E}_{z \sim p_z}[\log(1 - D(G(z)))] \qquad (7)$$

where $x \sim p_{data}$ represents the input subject to the real note distribution, and $z \sim p_z$ represents the analog distribution. $G$ is the generator, and $D$ is the discriminator. As is shown in Equation (7), by term $\mathbb{E}_{x \sim p_{data}}[\log D(x)]$ and $\mathbb{E}_{z \sim p_z}[\log(1 - D(G(z)))]$, the discriminator is expected to maximize the probability of sampled sequences being true and minimize the probability of generated sequences being false. Apparently, the output of the discriminator is the probability of being true music. Consequently, the only target label for $D(x)$ is 1, while for $D(G(z))$, it is 0. Therefore, Equation (7) can be seen as a variant of cross-entropy. When the generator $G$ is fixed, the partial derivative of the objective function $V(G, D)$ yields formula (8) for discriminator $D$:

$$D^*(x) = \frac{p_g(x)}{p_g(x) + p_{data}(x)} \qquad (8)$$

Substituting the optimal discriminator in Formula (7) into Formula (8) causes the optimization goal to optimize the Jenson–Shannon divergence (JSD) of $p_g(x)$ and $p_{data}(x)$. When $p_g(x) = p_{data}(x)$, they reach Nash equilibrium. Currently, the discrimination probability of discriminator $D$ for actual samples or generated samples is 50%.

This process has two problems: gradient non-differentiability and mode collapse. The reason why the gradient is not differentiable is that GANs need to input the generated note sequence into the discriminator to determine authenticity, but the output of the generated

model is the probability distribution with dimension [*BatchSize*, *SeqLength*, *NumNotes*]. The operation of argmax needs to be applied to each one-dimensional probability distribution to obtain the index of the note with the maximum probability, such that the generated sequence and real target have compatible and comparable shapes. However, as the function argmax returns the indices of the maximum values along an axis, the operation of argmax is nondifferentiable, which leads to a failure to solve the gradient between the judgment result of the discriminator and the trainable variables of the generator during the training step of the generator.

To make the process derivable, our solution to this problem is as follows: Firstly, the real sampling input sequence is transformed from an ID sequence into the sparse vector representation of the one-hot encoding, making the dimension of the real data and the generated probability distribution unified. Secondly, the output structure of the transformer is also optimized, and the inverse temperature parameter $\tau$ is introduced. The output optimization formula of the layer softmax is Equation (9):

$$y_i = \frac{\exp(y_i/\tau)}{\sum_{j=1}^{k} \exp(y_j/\tau)} \tag{9}$$

where $y_i$ is the normalized output probability of the $i$th note, $\tau$ is the inverse temperature parameter, $K$ is a global scalar, and $\sum_{j=1}^{k} \exp(y_j/\tau)$ is the normalizing term. As $\tau \to 0$, the probability distribution after Equation (9) approaches the one-hot encoding vector. As $\tau \to +\infty$, the output becomes a uniform probability. When $\tau$ is a finite positive value, the sample produced by Equation (9) is smooth and differentiable by the generator. To conclude, the relationship between probability distribution and extraction of the maximum value is expected to be learned by the inverse temperature parameter. During training, $\tau$ is set to a large value, which slowly decreases almost to zero.

The standard one-hot embedding represents the real sequence, and the generator's output is the probability distribution of the predicted label. There are huge differences in the expression form between the two, which the discriminator in GANs captures. Therefore, the one-hot embedding of the real sequence needs to be optimized, and noise added, as shown in Equation (10):

$$y_i = softmax\left(\frac{onehot(y_i) + g_i}{\lambda}\right) \tag{10}$$

where $y_i$ is the sequence of the real notes, $g_i$ is the random noise in the section $(-\varepsilon, \varepsilon)$, and $\lambda$ is a constant less than 1, which is used to amplify the result of noise. The purpose of adding $\lambda$ is to make the vector after softmax conversion closer to the form of one-hot encoding.

Another problem is mode collapse. The method used to evaluate the generation effect of the generator is to calculate the accuracy between the generated result and the actual result, but the system will mistake the one-hot encoding feature of the real sequence as one of the real features, resulting in the low accuracy of the generated model, and finally leading to mode collapse.

A root mean square error (RMSE) is added between the predicted and real sequence to accelerate the convergence and avoid mode collapse. When the difference between the predicted label and the real label exceeds a reasonable value, RMSE can correct the learning direction of the gradient.

Assuming the real target sequence sample $\{x_1, \ldots x_K\}$ and the real input sample $\{z_1, \ldots z_K\}$, the calculation formula of the loss function is as follows:

$$L = \frac{1}{K} \sum_{n=1}^{K} [logD(x_i) + \log(1 - D(G(z_i)))] \tag{11}$$

where $G$ is the generator, $D$ is the discriminator, $Z$ is a real input sample, and $n$ and $i$ are constants.

Therefore, the calculation formula of the loss function $L$ is as shown in Equation (12):

$$L = \alpha \frac{1}{K} \sum_{j=1}^{K} \| \hat{y}_j - G(z_j) \|^2 + \exp(\frac{1}{K} \sum_{n=1}^{K} \log(1 - D(G(z_i)))) \tag{12}$$

where $\alpha$ ($0 < \alpha < 1$) is the preset weight coefficient, $\hat{y}_j$ represents the real sequence label of the $j$-th sentence, and the former is the RMSE between the predicted sequence and the real sequence.

The calculation steps of the algorithm are as follows (Algorithm 1):

---

**Algorithm 1:** The Algorithm of Transformer and GANs

---

Input: real input sequence $\{z_1, \ldots z_n\}$, the target output sequence $\{x_1, \ldots x_n\}$
Output: predictive output sequence $\{y_1, \ldots y_n\}$
Random initialization generator and discriminator's parameters
**For** $i$ in $k$ steps to train do
　　**For** batch to iterate do
　　Discriminator training

- Input the real input sequence $\{z_1, \ldots z_n\}$ in $G$, generated $\{y_1, \ldots y_n\}$
- Input the false sequence $\{y_1, \ldots y_n\}$ and the target output sequence $\{x_1, \ldots x_n\}$ in $D$
- Used the descent gradient method to update the parameters, minimizing the loss

　　Generator training

- Input the real input sequence $\{z_1, \ldots z_n\}$ and the target output sequence $\{x_1, \ldots x_n\}$ in generator $\{y_1, \ldots y_n\}$
- Input the generated sequence in $D$, judge it as real or false
- Use the gradient descent method to update the parameters for $G$, minimize the combo loss

　　**End for**
**End for**

---

## 4. Experimental Summary

The GiantMIDI-Piano dataset, published by Jin et al. in 2020, was used to develop the proposed method [31]. Table 1 compares several major MIDI format music datasets. The GiantMIDI-Piano dataset is dramatically improved in quantity and richness compared to the others. More than 10,000 piano pieces with a total time of more than 1200 h can be played by algorithms. It is the most extensive classical piano dataset in the world.

**Table 1.** Piano dataset comparison.

| Dataset | Composers | Pieces | Hours | Types |
|---|---|---|---|---|
| Piano-midi.de | 26 | 571 | 36.7 | Seq. |
| Classical archives | 133 | 856 | 46.3 | Seq. |
| Kunstderfuge | 598 | - | - | Seq. |
| MAESTRO | 62 | 529 | 84.3 | Perf. |
| MAPS | - | 270 | 18.6 | Perf. |
| GiantMIDI-Piano | 2786 | 10,854 | 1237 | Live |

In Equation (12), the former is the root mean square error, and the latter is the cross-entropy loss. The loss of the cross-entropy term results in exponential amplification, making the model converge faster during the gradient of the training process. In addition, $\alpha$ avoids the process of convergence of authenticity learning instability because of an excessive RMSE. The value of $\alpha$ is selected by the training accuracy after five epochs, and the results are shown in Table 2. Note that since $\alpha$ is a hyperparameter used during the training of the generator, the accuracy being compared here is the accuracy of the generator.

**Table 2.** Influence of different values of $\alpha$ on the accuracy of the model.

| $\alpha$ | Accuracy after Five Epochs | $\alpha$ | Accuracy after Five Epochs |
|---|---|---|---|
| 0.1 | 0.21 | 0.6 | 0.18 |
| 0.2 | 0.22 | 0.7 | 0.13 |
| 0.3 | 0.24 | 0.8 | 0.15 |
| 0.4 | 0.23 | 0.9 | 0.13 |
| 0.5 | 0.18 | 1.0 | / |

According to Table 2, the accuracy rate changes when different values are substituted. When $\alpha$ is equal to 0.3, the accuracy rate is the largest, so the value of $\alpha$ is 0.3 (the result rounded up to 1 decimal place).

The proposed music generation model based on transformer and GANs has two loss optimization functions, corresponding to the optimization update of discriminator $D$ and generator $G$ in the generation countermeasure network. For discriminator $D$, the output is only 0 or 1, and the accuracy rate is the ratio of the predicted number of correct tags to the number of all tags. At each time step of prediction, the generator solves a multi-classification problem with the label dimension of vocab_size. For the prediction of each unit, its output is the probability distribution of the unit label.

Figure 3a is the process of discriminator loss rate. The loss of the discriminator decreases rapidly at the beginning of training until it is finally stable. Figure 3b is the change process of discriminator accuracy. The accuracy of the discriminator also increases rapidly to about 50% after the beginning of training, reaching the optimum state. Figure 3c is the change process of the correctness of the verification set. The accuracy rate of the verification set rises during the training process and finally reaches about 90%.

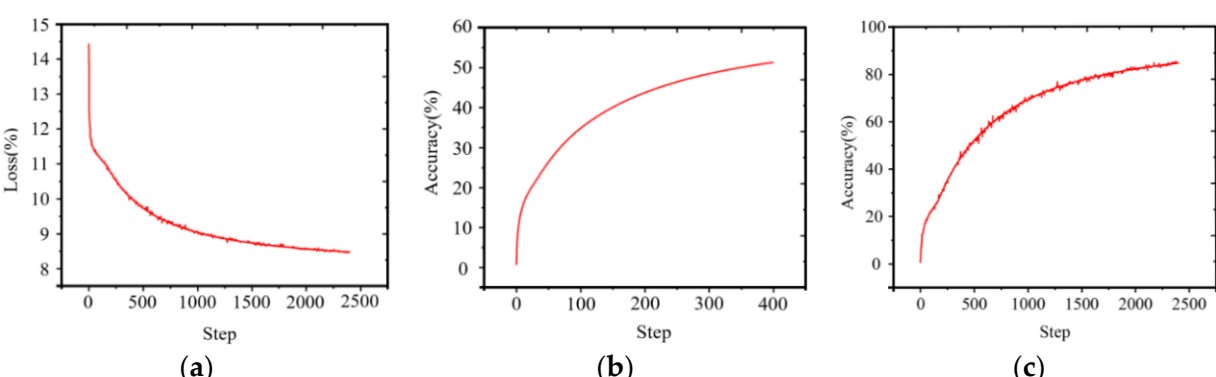

(a)         (b)         (c)

**Figure 3.** The result for the training process.

Figure 4a shows the input and output attention before optimization, while Figure 4b shows the input and output attention after optimization. Comparing Figure 4a,b, it can be seen that the image in Figure 4b is more complex than Figure 4a. At the same time, according to the visual experience, the color of Figure 4b is also darker than Figure 4a. Here, the image is used to reflect the relationship between the notes in generated melody and input samples. The color is darker, so the relationship between the input notes (or chords) is stronger.

In addition, the multihead attention mechanism in the transformer recognizes the relationship between input and output units. Figure 4a shows the input and output attention without using the optimization loss function, and Figure 4b shows the output attention after training with the optimization loss function.

The GAN model without the optimization function does not capture the relationship between input and output, but the optimized model learns the coupling relationship between input and output units far better. The final musical notation is shown in Figure 5.

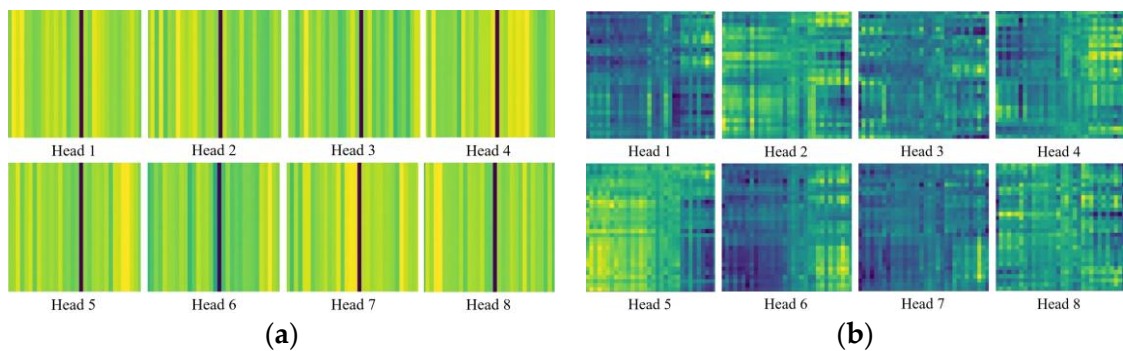

**Figure 4.** The input and output attention before optimization.

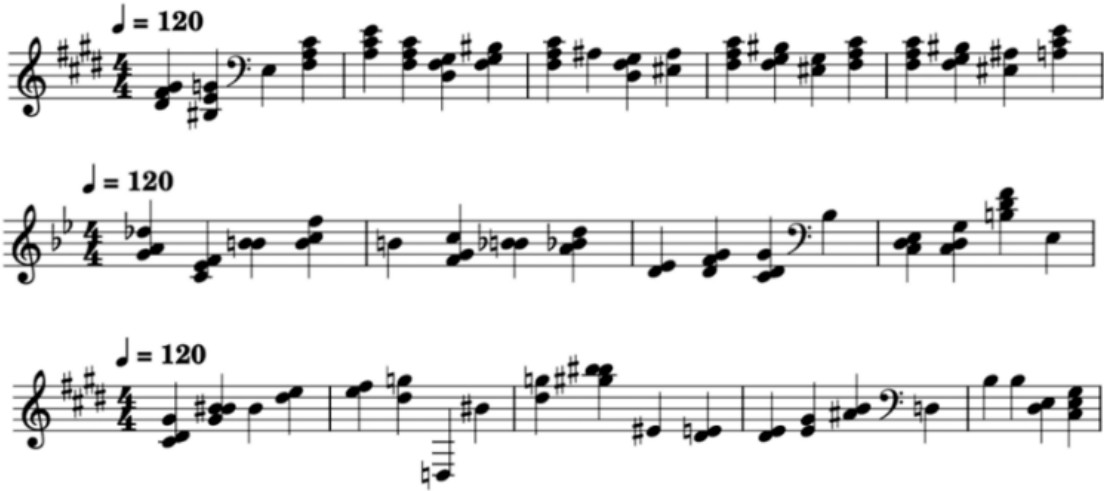

**Figure 5.** The results generated by the system based on transformer and GANs.

To make up for the defect of root mean square error in music evaluation, according to Euler's music evaluation elements, several commonly used elements in music evaluation, as shown in Table 3, were selected to quantify the advantages and disadvantages of the output music samples.

**Table 3.** The form for music evaluation.

| Parameter | Output Mode | Explain | Weigh $t$ |
|---|---|---|---|
| Absolute interval gradient | $w_1 * \frac{\sum_{i=0}^{n} s_g^i}{n-1}$ | $s_g^i$: Indicates the interval difference gradient score of the second note | 0.11 |
| Extreme note number | $w_2 * min\left(4 * \frac{n_{neibor}}{n}, 1\right)$ | $n_{neibor}$: Indicates the average number of extreme notes in the sequence | 0.29 |
| Dissonance | $w_3 * max\left(0, (-4) * \frac{n_{leap}}{n-1} + 1\right)$ | $n_{leap}$: Indicates the count value of a continuous jump | 0.17 |
| Chord monophonic ratio | $w_4 * \begin{cases} e^{10r_{chord}}, r_{chord} < 0.1 \\ 1, 0.1 \le r_{chord} < 0.5 \\ \ln(-r_{chord} + 1.5) + 1, r_{chord} \ge 0.5 \end{cases}$ | $r_{chord}$: Indicates the ratio of the number of chords | 0.29 |
| Note diversity | $w_5 * \left(-7r_{div}^2 + 7.5r_{div} - 0.75\right)$ | $r_{div}$: Indicates the ratio of the number of different notes to the length of the sequence | 0.14 |

**Note:** $n$: sequence length. $w_i$: output weight of each item. The weight for each parameter is just an example; the value can be adjusted appropriately according to preference.

The calculation formula of the absolute interval gradient is shown in Equation (13):

$$s_g = \begin{cases} 1, & x \leq 6 \\ \lg(-x + 16), & 6 < x \leq 15 \\ 0, & x > 15 \end{cases} \tag{13}$$

where $x$ represents the interval difference between notes.

The calculation formula for the number of extreme notes is shown in Equation (14):

$$n_{neibor} = \frac{2 * n_{min} * n_{max}}{n_{min} + n_{max}} \tag{14}$$

where $n_{min}$ indicates the number of notes in the bass in the presence of extreme note differences; $n_{max}$ indicates the number of notes in the range of treble; and $n_{neibor}$ indicates the average number of extreme notes.

The calculation formula for dissonance is shown in Equation (15):

$$n_{leap} = n_{leap} + 1, \; if \; abs(p_i - p_{i-1}) \& \; abs(p_{i+1} - p_i) \tag{15}$$

where $n_{leap}$ indicates the number of dissonances, and $p_i$ is the tone of the $i$ note.

The calculation formula for chord single tone ratio is shown in Equation (16):

$$r_{chord} = \frac{n_{chord}}{n_{chord} + n_{note}} \tag{16}$$

where $n_{chord}$ indicates the number of chords, and $n_{note}$ is the number of single notes.

The calculation formula for note diversity is shown in Equation (17):

$$r_{div} = \frac{n_{dif}}{n} \tag{17}$$

where $n_{dif}$ indicates the number of non-repeated notes, and $n$ is the total length of the sequence. The final score output is shown in Equation (18):

$$s = \frac{1}{\sum_{i=1}^{5} w_i} \sum_{i=1}^{5} w_i s_i \tag{18}$$

where $w_i$ is the output's weight for each item, $s_i$ is the output's value for each item, and $S$ is the final score.

The same sample dataset (GiantMIDI-Piano) [31] was input into several systems, as shown in Tables 4 and 5, to compare output results (full score is 100).

**Table 4.** Comparison of model results.

| Model | Original Sample (Average) | Output Melody (Average) | Training Accuracy |
|---|---|---|---|
| Long short-term memory (LSTM) | 83.30 | 46.62 | \ |
| Bidirectional long short-term memory (BiLSTM) | 85.75 | 55.83 | \ |
| Transformer and GANs (original loss) | 92.60 | 74.51 | 0.034 |
| Transformer and GANs (optimized loss) | | 92.33 | 0.562 |

Finally, we selected 30 volunteers with musical backgrounds from the Shanghai Conservatory of Music and 30 volunteers from the College of Electronics and Information Engineering at Tongji University for the test. Suppose that the volunteers from the Shanghai Conservatory of Music are professional and those from Tongji University are nonprofessional in the field of music. Based on Table 3, the total score is 100 points, and the scoring results are shown in Table 5. The results in Table 4 are calculated according to the elements involved in Table 3, and the calculation process refers to Equations (13)–(18). Table 5 is the

result of the volunteers' manual evaluation according to the elements involved in Table 3. The original samples for Tables 4 and 5 all used melodies included in the same sample dataset (GiantMIDI-Piano).

**Table 5.** Result of user study.

| Volunteer | Model | Original Sample (Average) | Output Melody (Average) |
|---|---|---|---|
| professional | Long short-term memory (LSTM) | | 42.32 |
| | Bidirectional long short-term memory (BiLSTM) | | 51.44 |
| | Transformer and GANs (original loss) | 94.07 | 65.71 |
| | Transformer and GANs (optimized loss) | | 82.58 |
| | Long short-term memory (LSTM) | | 63.65 |
| nonprofessional | Bidirectional long short-term memory (BiLSTM) | | 68.11 |
| | Transformer and GANs (original loss) | 92.82 | 75.01 |
| | Transformer and GANs (optimized loss) | | 88.33 |

Combining Tables 4 and 5, it can be seen that, for both data evaluation based on Euler's music evaluation elements or manual evaluation, the optimized transformer and GANs model has the highest scores. At the same time, compared with other models, the optimized transformer and GANs model also has the best accuracy for the notes.

## 5. Conclusions

Taking aim at the challenge of music generation, this study overcame the obstacles facing the sequence generation model and GANs, proposed a music generation model based on transformer and GANs, and optimized the structure of GANs. Through experimentation, it was found that the chord processing reported in this study is not ideal, as reflected in the high proportion of chords and excessive discordant notes. A character dictionary processed by this method was constructed according to the notes and chords in real music. To reduce the size of the dictionary and improve the prediction effect, word frequency was used as the basis for filtering, and some notes and chords with low frequency, extreme notes, and complex chords were shielded. Still, the number of chord labels is much larger than the number of individual notes. Although the number of single notes in the training data is much larger than the occurrence frequency of chords, the transformer is not sensitive to the frequency in the calculation process of the model, regardless of the relative probability of single notes and chords. The dataset selected in this study is based on piano compositions that contain many different chords and notes. Using only major and minor chords is not enough to express the music in this dataset. Therefore, preprocessing of music datasets and use of algorithms to effectively summarize the rules for the occurrence of notes should be the primary goals of music creation in the future.

**Author Contributions:** Conceptualization, J.M. and L.W.; methodology, Z.L.; software, D.L.; validation, M.Z., J.M. and L.W.; formal analysis, Z.L. and J.M.; investigation, D.L. and M.Z.; resources, J.M. and D.L.; data curation, J.M.; writing—original draft preparation, J.M. and L.W.; writing—review and editing, D.L., Y.H. and M.Z.; visualization, Y.H.; supervision, Y.H.; project administration, J.M. and L.W.; funding acquisition, Y.H. All authors have read and agreed to the published version of the manuscript.

**Funding:** This work was funded by Science and Technology Winter Olympic Project (Grant number 2018YFF0300505) and Joint Fund of Zhejiang Provincial Natural Science Foundation (Grant number LHY20F030001).

**Institutional Review Board Statement:** The study did not require ethical approval.

**Informed Consent Statement:** Not applicable.

**Data Availability Statement:** Not applicable.

**Conflicts of Interest:** The authors declare no conflict of interest.

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
