# Peer review of "Music Generation System for Adversarial Training Based on Deep Learning"

_processes, doi:10.3390/pr10122515_

Round 1

Reviewer 1 Report

## Summary

This work addresses the problem of computer music generation. It introduces a deep neural network architecture combining the transformer and a generative adversarial network.

The paper is divided into 5 sections.

## Detailed analysis

### Section 1: introduction

The introduction gives a very brief historical overview of computer-based music generation, before talking about DNNs in general and presenting some techniques such as CNNs, RNNs, generative models and the concept of embedding. However, the goal of the paper is not really well defined, since the objectives seem to be very ambitious and stated as "Generating structured music, analyzing the quality of the generated music, and building an interaction model are the extended problems that need to be solved". The contribution is mainly to combine transformer and GANs for this application, although the authors do not mention similar works such as Aashiq Muhamed et al. "Symbolic Music Generation with Transformer-GANs", 35th AAAI Conference on Artificial Intelligence, {AAAI} 2021. 

### Section 2: related works

The section highlights the importance of LSTMs in music generation and transformers in Natural Language Processing.

- The statement "Long short-term memory (LSTM) developed rapidly and has the most extensive application in music processing and the music generation field" is very questionable and is not supported by any references. LSTM is widely used and popular in many other fields besides music processing. Do the authors rather mean that LSTM is predominant in the field of music generation?

- "neural networks could not handle long-term sequences in music": why? If this is the case, then why teh authors propose a neural network-based solution for this problem?

### Section3: proposed method

The section 3 gives details of the proposed solution. It consist of a note conversion, an encoder with a multi-head attention mechanism, and a decoder. Special attention is given to the description of the adversarial loss amended with a MSE loss for training the generator in order to avoid mode collapse. No GAN advances like Wasserstein GAN or Least Squares GAN are mentioned or considered. The GAN training procedure given in pseudo-code follows the common way of doing. 

- Figure 2: "real or fake" decision block outputs are "no" and "yes", which are not clear and logical answers to the decision question. 

- "So, the value of argmax is assigned a subscript onehot forcibly, and then this subscript is  changed into a distribution so that the changes of x1 and x2 can transfer the changes to the values of the function argmax." is the solution from the author to the non-differentiability of the argmax function. However, the description is very difficult to follow and so imprecise that the reader can only guess what is really done. In general, there is a lot of approximation and confusion with the terms gradient and differentiability in the paper (the gradient is a value associated to a point of a differentiable function representing its variability).

### Section 4: Experimental summary

 The training data are not given in a precise way, as well as the number of parameters of teh model, the main hyperparameters (batch size, optimizer...) and the complexity estimation (the complexity advantage was given as justification for the choice of GAN in the introduction).

- Table 1 is not really relevant since only GiantMidi-Piano is used in the work.

- Table 2 reports the accuracy which is not clearly defined at this stage.

- The score in Table 4 is not defined. There are no details on how the other models were derived or designed (*implemented by the authors or from a publically available code).

- Table 5: There is no explanation of how the subjective assessment was conducted. What was the question asked to the listeners? Did the authors follow a known recommendation? If not, the procedure must be described in detail in order to let teh reader appreciate the results.

## General obserbvations:

- The level of detail and description does not allow readers to reproduce the results. 

- Several unclear and incomplete explanations hinder the quality of the document. 

- Some relevant references may be missing (see above). 

- The experimental section is too brief and imprecise to be convincing. 

- A demo would be welcome.

## Comments to the authors:

- L.41 p 1: "... apply deep neural networks to creating music." -> "... apply deep neural networks for creating music." (grammatical error)

- Introduction is not very informative. Sentences like "Some deep learning methods are autoregressive", w/o references are almost useless.

- "Google established the deep mind artificial intelligence laboratory in London to develop WaveNet":  is it really interesting for the scientific reader? Moreover it is not 100% correct, since DeepMind was already established before Google bought it..

- L261 p6 "model collapse" -> "mode collapse"

- L342, p : The figure is not properly inserted into the text.

Author Response

Thank you for your letter and for the reviewers’ comments concerning our manuscript entitled “Music Generation System for Adversarial Training Based on Deep Learning” (ID: processes-1990531). Those comments are all valuable and very helpful for revising and improving our paper, as well as the important guiding significance to our research. We have studied the comments carefully and have made corrections which we hope meet with approval. The main corrections in the paper and the responses to the reviewer’s comments are as flowing:

Point 1:

- The statement "Long short-term memory (LSTM) developed rapidly and has the most extensive application in music processing and the music generation field" is very questionable and is not supported by any references. LSTM is widely used and popular in many other fields besides music processing. Do the authors rather mean that LSTM is predominant in the field of music generation?

- "neural networks could not handle long-term sequences in music": why? If this is the case, then why do teh authors propose a neural network-based solution for this problem?

Response 1: We have modified the controversial expression. At the same time, LSTM cannot be used to deal with long sequence problems alone. Therefore, LSTM is generally combined with other neural networks to make the final model that can be used to solve long sequence problems.

Point 2:

- Table 1 is not really relevant since only GiantMidi-Piano is used in the work.

- Table 2 reports the accuracy which is not clearly defined at this stage.

- The score in Table 4 is not defined. There are no details on how the other models were derived or designed (*implemented by the authors or from a publically available code).

- Table 5: There is no explanation of how the subjective assessment was conducted. What was the question asked to the listeners? Did the authors follow a known recommendation? If not, the procedure must be described in detail in order to let teh reader appreciate the results.

Response 2: Table 1 compares several major MIDI format music datasets. Through Table 1, we can see the GiantMIDI-Piano dataset dramatically improved in quantity and richness more than the others.

For Table 2 and Table 4, we have added descriptions.

At the same time, the results in Table 4 are calculated according to the elements which involved in Table 3, and the calculation process refers to Formula 13-18.

The results in Table 5 are obtained from the questionnaire, and the volunteers make subjective evaluations according to the elements mentioned in Table 3.

Point 3:

- L.41 p 1: "... apply deep neural networks to creating music." -> "... apply deep neural networks for creating music." (grammatical error)

- L261 p6 "model collapse" -> "mode collapse"

Response 3: We have completed the modification of the above sentences and words

Reviewer 2 Report

This paper presents a novel music generation method based on using Transformer and GAN, which are both state-of-the-art effective deep neural network structures. A comprehensive review about the related works are provided, which is very helpful for potential readers. In addition, the presented method is described very clearly with reasonable explanations, and the experimental results are quite convincing and supportive. Based on these observations, I recommend acceptance. Some minor points have to be addressed in the final version:

1. These exist some grammatical errors the need correction.

2. In Eq. (8), the term D^{*} is not defined explicitly. Is it just the discriminator D?

3. The condition in Eq. (15) is not complete. 

Author Response

Thank you for your letter and for the reviewers’ comments concerning our manuscript entitled “Music Generation System for Adversarial Training Based on Deep Learning” (ID: processes-1990531). Those comments are all valuable and very helpful for revising and improving our paper, as well as the important guiding significance to our research. We have studied the comments carefully and have made corrections which we hope meet with approval. The main corrections in the paper and the responses to the reviewer’s comments are as flowing:

Point 1: There exist some grammatical errors that need correction.

Response 1: We have completed the modification of the above sentences and words

Point 2: In Eq. (8), the term D^{x} is not defined explicitly. Is it just the discriminator D?

Response 2: In Eq. (8), D^{x} is the result for discriminator D when the input for the system is x.

Reviewer 3 Report

In this article, the authors present the influence of AI on music generation. GANs are getting more and more popularity to generate music sequences and the authors offer a new model that combines GANs and a Transformer.  

The article is well-structured and the aim of the article is clear. There is the introduction to the topic and a presentation of related works. Then, the authors depict the method used to lead their research with the outcomes received. The method is demonstrated with figures and formulae to facilitate the reception of the text. The layout of the whole article is well-considered. 

The references are relevant and contain both the older works related to the beginnings of AI and computer-based music generation and the latest ones concerning the state of the current research and possibilities. 

Figure 2 – leave the title of the figure under the picture (The structure of the music generation system). The whole explanation should be incorporated into the text under the picture, not in the title of the figure. 

Author Response

Thank you for your letter and for the reviewers’ comments concerning our manuscript entitled “Music Generation System for Adversarial Training Based on Deep Learning” (ID: processes-1990531). Those comments are all valuable and very helpful for revising and improving our paper, as well as the important guiding significance to our researches. We have studied comments carefully and have made correction which we hope meet with approval. The main corrections in the paper and the responds to the reviewer’s comments are as flowing:

Point : Figure 2 – leave the title of the figure under the picture (The structure of the music generation system). The whole explanation should be incorporated into the text under the picture, not in the title of the figure.

Response : We have completed the modification of Figure 2.

Round 2

Reviewer 1 Report

## General

The authors made minor corrections to the manuscript and did not address the main issues highlighted in the first review.

For example: In Figure 2 the decision "true or false?" is still answered by "No and Yes". Another point is related to the many confusions between gradient and derivability, which was not addressed as well. Brough corrections are also not always done in the right way, e.g. line 103 leading to a grammatically incorrect sentence.

## Non-exhaustive list of additional technical issues 

Several loose descriptions raise doubts about the technical soundness. For example, in equation (7), the GAN optimization function differs from the standard minimax loss function, where the second term is 1-log(D(G(z))) instead of log(1-D(G(z))). This incorrect or new formulation is repeated in the text. This term is also called "cross-entropy", in the whole text, although it is not a comparison between two distributions. By contrast, the whole equation (7) derives from the cross-entropy between the real and generated distributions. In the same section the authors seems to have use the term "heat coding" instead of "one-hot encoding" adding even more confusion.

## Regarding the experimental section

The new version still not define precisely what is the accuracy at the time Table 2 is introduced. It leads then to questions of this kind:  Is the accuracy of Table 2 and the discriminator accuracy of Fig. 4 the same? 

By reading the text "Based on Table 3, the total score is 100 points, and the scoring

results are shown in Table 5.", one will deduce that Table 5 is derived from equation (18), since Table 3 lists all parameters used in eq (18). However, authors response to report 1 states the opposite: "**At the same time, the results in Table 4 are calculated according to the elements which involved in Table 3, and the calculation process refers to Formula 13-18.**

**The results in Table 5 are obtained from the questionnaire, and the volunteers make subjective evaluations according to the elements mentioned in Table 3.**", which seems to make a lot of sense but which is contracted in the text.

Nonetheless, the exact procedure used for the subjective evaluation is not given which is a major flaw (for instance questions like the following are not answered:  duration of the test/number of evaluated sequences, did the volunteers evaluate the real note sequences as well?....).

Author Response

请参阅附件。
